# Accurate Detection of SARS-CoV-2 by Next-Generation Sequencing in Low Viral Load Specimens

**DOI:** 10.3390/ijms24043478

**Published:** 2023-02-09

**Authors:** Marius Ilié, Jonathan Benzaquen, Véronique Hofman, Elodie Long-Mira, Sandra Lassalle, Jacques Boutros, Christophe Bontoux, Virginie Lespinet-Fabre, Olivier Bordone, Virginie Tanga, Maryline Allegra, Myriam Salah, Julien Fayada, Sylvie Leroy, Matteo Vassallo, Irit Touitou, Johan Courjon, Julie Contenti, Michel Carles, Charles-Hugo Marquette, Paul Hofman

**Affiliations:** 1Laboratory of Clinical and Experimental Pathology, Centre Hospitalier Universitaire de Nice, FHU OncoAge, Université Côte d’Azur, 06000 Nice, France; 2Hospital-Related Biobank (BB-0033-00025), Centre Hospitalier Universitaire de Nice, FHU OncoAge, Université Côte d’Azur, 06000 Nice, France; 3Team 4, Institute of Research on Cancer and Aging (IRCAN), CNRS INSERM, Université Côte d’Azur, 06107 Nice, France; 4Department of Pulmonary Medicine and Oncology, Centre Hospitalier Universitaire de Nice, FHU OncoAge, Université Côte d’Azur, 06000 Nice, France; 5Department of Internal Medicine and Oncology, Centre Hospitalier de Cannes, 06400 Cannes, France; 6Department of Infectious Diseases, Hôpital Archet 1, Centre Hospitalier Universitaire de Nice, Université Côte d’Azur, 06200 Nice, France; 7Emergency Department, Hôpital Pasteur 2, Centre Hospitalier Universitaire de Nice, Université Côte d’Azur, 06000 Nice, France

**Keywords:** COVID-19, SARS-CoV-2, low viral load, next generation sequencing, Genexus

## Abstract

As new SARS-CoV-2 variants emerge, there is an urgent need to increase the efficiency and availability of viral genome sequencing, notably to detect the lineage in samples with a low viral load. SARS-CoV-2 genome next-generation sequencing (NGS) was performed retrospectively in a single center on 175 positive samples from individuals. An automated workflow used the Ion AmpliSeq SARS-CoV-2 Insight Research Assay on the Genexus Sequencer. All samples were collected in the metropolitan area of the city of Nice (France) over a period of 32 weeks (from 19 July 2021 to 11 February 2022). In total, 76% of cases were identified with a low viral load (Ct ≥ 32, and ≤200 copies/µL). The NGS analysis was successful in 91% of cases, among which 57% of cases harbored the Delta variant, and 34% the Omicron BA.1.1 variant. Only 9% of cases had unreadable sequences. There was no significant difference in the viral load in patients infected with the Omicron variant compared to the Delta variant (Ct values, *p* = 0.0507; copy number, *p* = 0.252). We show that the NGS analysis of the SARS-CoV-2 genome provides reliable detection of the Delta and Omicron SARS-CoV-2 variants in low viral load samples.

## 1. Introduction

Control of the COVID-19 pandemic requires tremendous effort to rapidly and massively screen and detect known or emerging variants of SARS-CoV-2 [1]. The emergence of these new variants, initially unknown or not detected in certain countries, can result in the rapid progression of infected clusters of individuals [2,3,4,5].

Since the beginning of 2022, the Omicron variant of SARS-CoV-2 has increased worldwide and has spread more rapidly than the Delta variant [6]. Importantly, RT-PCR Ct values seem to be higher for Omicron than for Delta regardless of the vaccination status, suggesting that Omicron is associated with a lower viral load compared to the Delta variant [7]. In addition, screening by full genome sequencing of SARS-CoV-2 is performed in only 1.5% of positive cases, depending on the national recommendations, which may lead to a bias in controlling genomic surveillance, in particular when dealing with low viral loads [8,9].

In this context, reliable and accurate laboratory detection methods for SARS-CoV-2 play a critical role in genomic surveillance and in controlling viral transmission. Screening tests for SARS-CoV-2 detection and characterization using RT-PCR demonstrate a relatively low sensitivity leading to some false-negative results, which may be related to a low viral load in the nasopharyngeal secretions of some patients [7,10]. The specificity of most of the RT-PCR tests is 100% because the primer design is specific to the genome sequence of SARS-CoV-2. Occasional false-positive results may occur due to technical errors and reagent contamination.

Consequently, a more discriminant approach to screening the different variants of SARS-CoV-2 in the population by genomic sequencing is crucial to monitor the emergence of new variants as well as their expansion in a given country or region [11,12,13]. In this regard, complete viral genome sequencing should be specific and sensitive enough to get robust results from specimens with low viral loads, even on specimens obtained in the community from asymptomatic patients, and should not be restricted to hospital samples obtained from patients with high viral loads.

We report here the performance of an NGS approach Genexus for SARS-CoV-2 genomic sequencing using specimens with low viral loads. For this study, we used samples from the fifth wave of the pandemic in the Southeastern region of France, thus providing data on the progression of the epidemic and Delta-to-Omicron ΒA.1 transition patterns.

## 2. Results

Since the end of October 2021 and notable in December 2021 an increase in hospitalization of patients with COVID-19 progressed slowly, and after a short decrease, a second spike was observed in February 2022 (Figure 1).

The second faster increase in the admission rate correlated with an increase in the detection of the Omicron variant that went from less than 1% of sequenced strains in mid-December to the dominant variant in less than 4 weeks (Figure 1). Patients were predominantly women (110/175, 63%) and the mean age (±SD) was 41 ±9 years. No differences were observed between patients infected with the Omicron or Delta variant for gender, age, and time since symptom onset.

The Ct N values ranged from 20.6 to 43.6 (median ±SD, 35.3 ±4.44). Among the 175 positive samples, only three (1.7%) were strongly positive (Ct < 24), 38 (21.7%) were moderately positive (24 ≤ Ct < 32), and 134 (76.5%) were weakly positive (Ct ≥ 32).

The quantity of viral RNA varied from 1 to 230,400 copies/µL (median = 15 copies/µL), with 133 (76%) cases having a low viral load of ≤ 200 copies/µL.

The NGS analysis was successful in 159 (91%) cases. The lineage analysis showed that 100 (57%) cases harbored the Delta variant, and 59 (34%) were identified with the Omicron BA.1.1 variant. After repeated sequencing runs, complete failure to obtain a valid sequencing result was observed in 16/175 (9%) of the samples. Failure was certainly due to degraded RNA combined with a low quantity of viral RNA (median, 20 copies/µL).

No significant difference was observed between the Ct values and the sequencing depth (*p*-value = 0.236). Samples showed mapped reads with an average depth ranging from 1329 to 26,772 (median, 2149).

Finally, there was no significantly lower viral load in patients infected with the Omicron variant compared to the Delta variant (Figure 2).

## 3. Discussion

We demonstrated the feasibility of using the Ion AmpliSeq SARS-CoV-2 Insight Research Assay on the Genexus platform to identify Delta and Omicron SARS-CoV-2 variants in samples with low viral loads. This approach supports the monitoring of the disease’s spread and activity and evolution of the virus [14]. Since the beginning of the COVID-19 pandemic at the end of 2019, numerous SARS-CoV-2 variants have emerged in different countries and have become disseminated in different regions of the world [15]. The identification and detection of all the variants highlight the urgent need to develop large genomic sequencing programs in all countries to be able to immediately identify the onset of a potential new variant having high infectiousness and possible resistance to the different types of immune responses developed by current vaccines [15]. The detection of SARS-CoV-2 using RT-PCR can demonstrate high variability of sensitivity in various diagnostic specimens (e.g.,., bronchoalveolar lavage, double naso/oropharyngeal swabs, nasopharyngeal swabs, saliva, and oropharyngeal swabs) as reported elsewhere [16]. A high Ct PCR, i.e., a very low viral load on a specific sample does not systematically predict the absence of an active viral infection. Thus, being able to identify a viral strain on a not-very-sensitive specimen can be of interest to some patients, at risk of severe SARS-CoV-2 infection, and even more, if new variants are emerging. In the era of active treatments such as antivirals and monoclonal antibodies that have a variable efficiency according to the viral strains, being able to identify any variant could have clinical consequences in terms of treatment strategies. Additionally, our findings extend far beyond local problems and are significant for epidemiological purposes.

SARS-CoV-2 genome sequencing, or at least complete or partial spike (S)-gene sequencing, seems the best method to characterize a specific variant [8]. Alternative methods, such as RT-PCR-based assays, have been developed for early detection and pre-screening to allow prevalence calculation of variants of concern (VOCs), variants of interest (VOI), and variants under monitoring (VUM). When these assays are used, confirmatory sequencing should be performed to use the results as indicators of community circulation of virus variants, particularly VOCs. However, full viral sequencing is performed only in a fraction of positive cases, which leads to a bias in genomic surveillance [8,17].

By the end of 2021, a new variant named Omicron emerged and was immediately classified as VOC [18]. The Omicron variant showed high transmissibility and quickly displaced the Delta variant, becoming dominant worldwide in only a few weeks [19]. In our study, the Omicron variant outcompeted the previously dominant Delta variant in only four weeks. While higher transmission rates have been related to higher viral loads of the Delta variant, recent studies showed lower viral loads for patients infected with Omicron, compared to patients infected with the Delta variant [7,10,20]. Therefore, the lower viral loads of Omicron infections could affect its epidemiological surveillance when using less specific screening tests. Higher Ct values in Omicron cases may be associated with an increased number of false negative results compared to Delta cases when analyzed by RT-PCR. Moreover, determining the cut-off of positivity of samples with a low viral load versus being called indeterminate or equivocal could engender false-positive results, which could jeopardize a screening strategy based on methods with a low specificity [21,22].

Here, we analyzed by NGS 23 specimens identified as negative with Idylla RT-PCR SARS-CoV-2. Among these, 39% (9/23) were negative with NGS; however, 61% (14/23) were positive for SARS-CoV-2 giving a high rate of false-negative results. This preliminary result must not preclude the use of RT-PCR assays for SARS-CoV-2 screening, however, the interpretation must be cautious and new criteria must be defined as indicators of infectiousness.

Additionally, only approximately 50% of positive results for COVID-19 could be achieved because of the limits of RT-PCR technology, which were then, verified with chest CT and other diagnostic tools [23]. The timing of the test in relation to the acquisition of the virus is another issue that is addressed with this idea. When interpreting RT-PCR results, especially early in the course of an illness, special caution should be used [24,25]. Following the initial diagnosis, avoiding false-negative tests from convalescent patients who are going to be discharged and out of quarantine is crucial to prevent the risk of transmission and recurrence. Therefore, to ensure the accuracy of SARS-CoV-2 virus detection, sensitive procedures must be developed.

In our study, Delta and Omicron variant identification by NGS was efficient for low viral loads (Ct ≥ 32, ≤200 copies/µL). Our findings demonstrated the potential for SARS-CoV-2 genome sequencing from samples with low viral loads using the Ion Ampliseq SARS-CoV-2 Assay carried out on the Genexus platform. The failure rate was low (9%) and was most probably due to degraded RNA combined with very low viral loads.

However, since only a maximum of 15 samples can be processed daily in a routine clinical practice with the Genexus automated system, the flow of sequencing is limited compared to the large flow of screening using RT-PCR methods. Setting up a strategy for quick integration of SARS-CoV-2 NGS after variant screening using multiplex RT-PCR may therefore be beneficial. RT-PCR panel tests for SARS-CoV-2 that are currently available are not totally sensitive, and therefore, they may miss some variants [26]. As a result, secondary genomic sequencing of nasopharyngeal and saliva samples that are positive for RT-PCR SARS-CoV-2 but lack comprehensive variant identification can be achieved [27]. However, this two-step analysis could potentially cause a delay in the final diagnosis and restrict the speed with which preventative health measures can be put in place to stop the spread of newly discovered variants with unknown or well-known infectiousness. In contrast, an upfront SARS-CoV-2 NGS analysis would enable simultaneous screening for new developing variants and prompt detection of all existing variations. Only in a few circumstances, most notably when there are no more than 15 samples to be processed per 24 h, is the latter alternative feasible. These situations involve the discovery of small clusters in places such as companies, schools, and retirement homes. Additionally, when SARS-CoV-2 cannot be identified using RT-PCR from nasopharyngeal samples, a few patients may exhibit clinical symptoms that are suggestive of COVID-19. SARS-CoV-2 NGS may therefore be performed right away using other biological sources (bronchial aspirates or biopsies, or broncho-alveolar lavages, etc.) [28].

The present study has some limitations. First, as this was a single-site retrospective study, the NGS approach should be validated in daily practice with the usual workflow of samples in a prospective manner. Secondly, only the Delta and Omicron BA.1 variants were circulating in France during the study period. Thus, we could not validate our NGS approach on other lineages [29]. Even though tiling amplicon designs try to reduce the consequences of such alterations, SNP or indels located with primer-annealing regions can still have an impact on amplicon-based target enrichment. Primers for amplicon-based target enrichment also need to be updated frequently since SARS-CoV-2 may mutate. Further studies could be performed to demonstrate that this approach may be of interest to identify unknown SARS-CoV-2 variants. Third, orthogonal methods have not been used in the present study to confirm the genomic data [30,31,32]. Finally, the NGS approach, similar to the RT-PCR assays, enables the identification of viral genomes at low viral loads but does not enable the determination of the viability of the virus. For instance, the likelihood of isolating a viable virus is thought to be between 0 and 15% for a Ct of 35 [33].

The Genexus sequencing has fewer gaps (<250 mean N per genome) than the Illumina and Oxford Nanopore Technology (ONT) platforms (>1250 mean N per genome) in a survey of 100,000 SARS-CoV-2 genome sequences [34]. Several research papers report high error rates in ONT sequencing (5–8%) [35,36,37]. The Genexus system offers complete integration from library preparation to analysis in a single workflow, whereas ONT and Illumina have multiple instruments and 3rd party requirements. The Illumina approach needs separate instruments for automated liquid handling, thermocycler, plate centrifuge, qPCR system, and sequencer [38], whereas the ONT approach has third party requirements for extraction, cDNA synthesis, and bioinformatics analysis and is a highly manual process [35]. In addition, the ONT system has the fastest turnaround time (~11 h), the Illumina platform has the longest turnaround time (3–4 days), while the Genexus system lasts less than 24 h. The Genexus platform has one touchpoint from library prep to analysis, the Illumina system requires a large amount of hands-on time, and ONT has many steps (Table 1) [39].

Finally, the Genexus system is an easy-to-use “plug and play” with no requirement for prior bioinformatics experience, and has an integrated software for sequencing analysis and reporting. The Illumina platform requires effort to automate, and the need to develop and verify scripts for automation for different liquid handlers. For instance, the DRAGEN COVID lineage analysis workflow is complex with known limitations [40].

In conclusion, our study demonstrated that an NGS approach with the Ion AmpliSeq SARS-CoV-2 Insight Research Assay on the Genexus platform was able to detect Delta and Omicron VOCs in samples with low viral loads. In this context, the SARS-CoV-2 NGS may be implemented as a screening approach, depending on the clinical presentation, the severity of the medical emergency, and the volume of samples collected on a daily basis.

## 4. Materials and Methods

### 4.1. Patients and Samples

This retrospective observational cohort study was conducted over a period of 32 weeks (from 19 July 2021 to 11 February 2022) on 175 patients treated for COVID-19 in the Departments of Pulmonary and Critical Care Medicine, Infectious Diseases and Emergency at the Nice University Hospital (Nice, France). Consecutive patients giving their consent were enrolled in the study. The initial COVID-19 detection before hospital admission was performed with commercialized antigen or RT-PCR tests performed outside the hospital. To ensure comparable Ct values for viral load analyses, samples were retested during hospitalization with the RT-PCR Idylla SARS-CoV-2 test (Biocartis, Mechelen, Belgium) [41].

All samples were stored at −80 °C at the University Côte d’Azur COVID-19 Biobank (BB-0033-00025, Pasteur Hospital, Nice, France) prior to their analysis [42]. The sponsor of the study was the Centre Hospitalier Universitaire de Nice (ClinicalTrial.gov identifier: NCT04418206). This study was approved by the CPP Sud Méditerranée V ethics committee (2020-A01050-39). All subjects signed an informed consent to participate in this work.

### 4.2. SARS-CoV-2 Detection and Genome Sequencing

All samples underwent one freeze-thaw cycle. Viral RNA was extracted with the MagMAX™ Viral/Pathogen Nucleic Acid Isolation Kit ((#A42352, Applied Biosystems, Foster City, CA, USA) using the KingFisher™ Duo Prime Purification System (Thermo Fisher Scientific, Waltham, MA, USA).

In order to prepare the normalization of RNA samples before sequencing, we measured the viral RNA copy number with TaqPath™ 1-Step RT-qPCR Master Mix (#A15299, Thermo Fisher Scientific) and TaqMan™ 2019nCoV Assay Kit v1 (#A47532, Thermo Fisher Scientific) using the 7500 Fast real-time PCR System (Applied Biosystems). For each 2019-nCoV assay, we combined the following components (MasterMix, RNAse P assay, and viral standards—ORF1ab protein or S protein or N protein. For each reaction, we combined the following components (nucleic acid research sample and 2019-nCoV human control standard and no template control). The copy numbers were calculated based on the Ct values, obtained by qPCR for the S, N, and ORF1ab targets, which were translated into the correspondence table in the quick reference “Ion Ampliseq SARS-CoV-2 research panel” (Thermo Fisher Scientific, Waltham, MA, USA) [1]. Quantitative RNA data can be seen in RT-qPCR (copy number) and Idylla (Ct viral load). The quality of the amplified RNAs can be checked during sequencing by analyzing the % N (undefined nucleotide). RT-qPCR experiments were performed in compliance with the MIQE guidelines [43].

Sequence analyses were performed on the Genexus platform (Thermo Fisher Scientific) using the Ion AmpliSeq SARS-CoV-2 Insight Research Assay—GX (Thermo Fisher Scientific), which includes two pools of 237 amplicons of 125 to 275 bp, covering the entire genome of SARS-CoV-2, as previously described [44]. For each run, 15 samples from the individuals and an internal control were used. The quality check of the sequencing was performed with the “Ion AmpliSeq SARS-CoV-2-LowTiter Research Assay (Thermo Fisher Scientific) specially designed for ≤200 copies. The fastq files were quality filtered and reads mapped with the SARS-CoV-2-Pangolin plugin (https://cov-lineages.org/resources/pangolin.html), the “COVID19AnnotateSnpEff” automatic plugin (Thermo Fisher Scientific), as well as the Nextclade tool (https://clades.nextstrain.org), against the reference genome from Wuhan (GenBank accession number NC_045512.2), to achieve the complete viral genome sequences. The sequencing data were deposited at the European Nucleotide Archive (project number PRJEB47330; https://www.ebi.ac.uk/ena/browser/view/PRJEB47330?show=xrefs, accessed on 25 November 2022).

The criteria to define a valid sequencing result or a complete failure were: (i) the number of reads higher than 1 million, (ii) <1% of unreadable nucleotides in the sequence, and (iii) the average depth higher than 1000, as previously shown.

### 4.3. Statistical Analysis

Continuous variables are presented as medians (±standard deviation and range); categorical variables are as numbers and percentages. Comparisons between variables two by two were evaluated using the unpaired Student’s t-test. Biological variables were tested with and without logarithm transformation. All statistical analyses were performed with StatAid Software at the significance threshold of 0.05 [45].

## Figures and Tables

**Figure 1 ijms-24-03478-f001:**
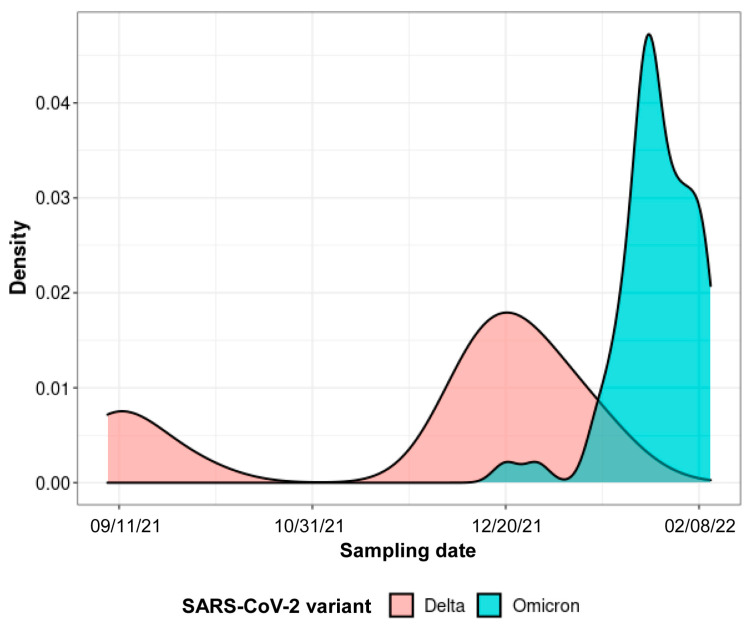
Dynamics of COVID-19-related hospital admissions and SARS-CoV-2 variant distribution between September 2021 and February 2022. Density chart representation.

**Figure 2 ijms-24-03478-f002:**
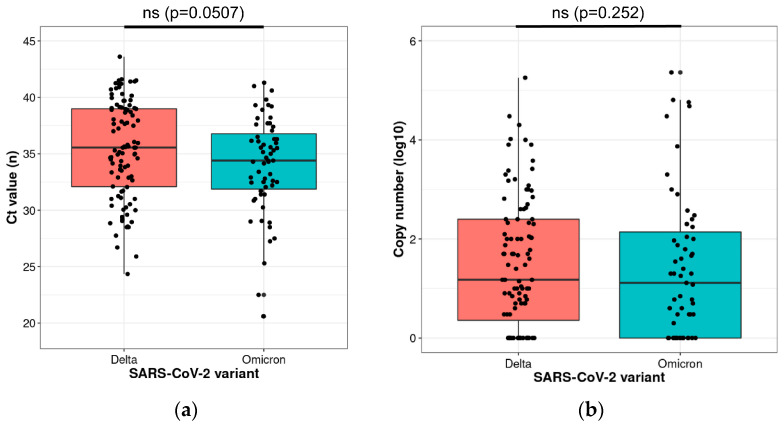
(**a**) Cycle Threshold (Ct) value of Idylla RT-PCR assay according to the SARS-CoV-2 variant; (**b**) Viral load according to SARS-CoV-2 variant. Delta, *n* = 100 samples; Omicron, *n* = 59 samples. Boxplot and scatter dot plot representation. Dots outside whiskers are considered outliers. ns = non-significant. Unpaired Student’s t-test.

**Table 1 ijms-24-03478-t001:** Comparison of different SARS-CoV-2 sequencing approaches.

	Genexus Integrated Sequencer	Illumina NextSeq RUO COVID-Seq	Midnight ONT
Workflow TAT	1 working day	3 working days	1 working day
Total run time	24 h	40 h	11 h
Hands-on time	5 min	3 h	1 h 10 min
Touchpoints	1	>80	>55

## Data Availability

The sequencing data were deposited at the European Nucleotide Archive (project number PRJEB47330; https://www.ebi.ac.uk/ena/browser/view/PRJEB47330?show=xrefs).

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
