# Peer review of "Accurate Detection of SARS-CoV-2 by Next-Generation Sequencing in Low Viral Load Specimens"

_ijms, 2023, doi:10.3390/ijms24043478_

Round 1

Reviewer 1 Report

Abstract needs some clarification. You stated that lineage analysis showed that 57% of cases harboured the delta variant  and 34% the omicron variant. Presumably this means 57% of cases for which NGS was successful and 34 % of cases for which NGS was successful? This needs to be explicitly stated. Moreover, 57% plus 34% = 91%, the same percentage as the percentage of samples for which NGS was successful- hence my desire for clarification. This leaves 9% with the variant  neither  delta nor omicron. Does this mean that the sequence was not readable , or the variant was not identifiable, or it was  a non-omicron, non-delta variant? This section of the abstract needs re-writing and clarification.  Comments on Materials and Methods, section 4.2, paragraph 3:Was the sequencing reaction carried out directly on the extracted, purified RNA, or was there a preampfification by RT-PCR then the amplicons subjected to sequencing on the Genes platform. I suspect the former was the case, but this should be stated explicitly. Section 4.2 paragraph 4. Rather than saying <1% "unknown" nucleotides (we know they are either U,A,T or G), it would be better to say <1% of "unreadable" or unidentifiable nucleotides. Section 4.3: say "significance threshold" not "significant threshold". Say "comparisons" between variables , not "links" between variables. Were internal standard sequence(s) used in the RT-qPCR for copy number quantification? What were these standards?  Did the RT-qPCR comply with all of the MIQE guidelines? This should be stated and the relevant paper cited. All qPCR papers must comply with MIQE guidelines. https://pubmed.ncbi.nlm.nih.gov/19246619/  Materials and methods should be made more detailed to demonstrate how this compliance was achieved.

Author Response

Reviewer #1:

Abstract needs some clarification. You stated that lineage analysis showed that 57% of cases harboured the delta variant and 34% the omicron variant. Presumably, this means 57% of cases for which NGS was successful and 34 % of cases for which NGS was successful? This needs to be explicitly stated. Moreover, 57% plus 34% = 91%, the same percentage as the percentage of samples for which NGS was successful- hence my desire for clarification.

Authors’ response: We thank the reviewer for these comments and we totally agree. We now clarified this in the abstract.

This leaves 9% with the variant neither delta nor omicron. Does this mean that the sequence was not readable, or the variant was not identifiable, or it was a non-omicron, non-delta variant? This section of the abstract needs re-writing and clarification.  

Authors’ response: The 9% of failed cases had unreadable sequences. We now clarified this point in the abstract. 

Comments on Materials and Methods,

section 4.2, paragraph 3: Was the sequencing reaction carried out directly on the extracted, purified RNA, or was there a preampfification by RT-PCR then the amplicons subjected to sequencing on the Genes platform. I suspect the former was the case, but this should be stated explicitly.

Authors’ response: We totally agree with the reviewer and we now clarified this point in the manuscript. The sequencing reaction was carried out directly on the extracted and purified RNA. The pre-amplification step by RT-PCR was used to quantify viral RNA copy number in order to prepare the normalization of RNA samples before sequencing.

Section 4.2 paragraph 4. Rather than saying <1% "unknown" nucleotides (we know they are either U,A,T or G), it would be better to say <1% of "unreadable" or unidentifiable nucleotides.

Authors’ response: We modified accordingly to <1% unreadable nucleotides.

Section 4.3: say "significance threshold" not "significant threshold". Say "comparisons" between variables, not "links" between variables.

Authors’ response: We totally agree with the reviewer and we modified accordingly both sentences.

Were internal standard sequence(s) used in the RT-qPCR for copy number quantification?

What were these standards?  

Authors’ response: We completed this section for more clarity now.

For each 2019-nCoV assay, we combined the following components (MasterMix, RNAse P assay, and viral standards – ORF1ab protein or S protein or N protein. For each reaction, we combined the following components (nucleic acid research sample and 2019-nCoV human control standard and no template control).

Did the RT-qPCR comply with all of the MIQE guidelines? This should be stated and the relevant paper cited. All qPCR papers must comply with MIQE guidelines. https://pubmed.ncbi.nlm.nih.gov/19246619/ Materials and methods should be made more detailed to demonstrate how this compliance was achieved.

Authors’ response: We thank the reviewer for these important comments. Therefore, we now detailed more the Materials and methods section in compliance with the MIQE guidelines. 

Reviewer 2 Report

The manuscript submitted by Ilie et al, describes the comparison of RT-qPCR with next generation sequencing pipeline using the Ion Torrent Genexus technology platform.  Whilst the platform represents a step forward in automated high-thoughput processing of clinical samples for whole genome sequencing of COVID-19 patient samples, it is my opinion that the study has some challenges in making this data novel enough for publication and more detailed analysis is required to increase its novelty.

1. From my point of view the main message of the manuscript was confusing - Is the focus here is to identify the Genexus pipeline as a candidate for a diagnostics pipeline in clinical laboratories or is it focusing on the 9% of samples that failed (and why they failed)?  If it is the former, then this should be clarified/confirmed in the text and data should be included on limit of sensitivity, inter/intra run variation and implications for difficult samples?  If it is the latter, more information regarding these samples (provenance/QC) should be included.

2. Was additional phylogenetic and variant analysis performed upon all samples other than those included in the pipeline?  Were omicron minority variants picked up earlier than the peak - if so, at what time and could these have been used to predict a new wave as argued in both the introduction and discussion?

3. How were these results validated – whilst I appreciate that the authors have stated that no orthogonal analysis was performed (L190) this significantly narrows the number of conclusions that can be made on this data?  What are the implications for contamination using this pipeline, for instance?

4. Other than simply Delta and Omicron outputs from the assay, were any additional data available to identify other novel variants.  As stated on L230 this pipeline is amplicon based and therefore is sensitive to novel variants and amplicon drop-outs/revisions.  How did that translate across all samples.  What % of genomes were identified in samples using this method?

5. How do these results relate to the clinical outcome of the cases?  Whilst the paper acknowledges its small sample set, how relevant are these results to the larger picture in France (rather than just the South eastern region)?

6. Relating to point 1, how did the authors identify samples as ‘certainly due to degraded and low viral load’ (L95)?  What were these observations based on?  This should be a minimal requirement for the manuscript as its main focus is on those samples that are low level/undetectable by other methods.  Could these be used to predict problem samples before sequencing? 

Author Response

Reviewer #2:

The manuscript submitted by Ilie et al, describes the comparison of RT-qPCR with next generation sequencing pipeline using the Ion Torrent Genexus technology platform.  Whilst the platform represents a step forward in automated high-thoughput processing of clinical samples for whole genome sequencing of COVID-19 patient samples, it is my opinion that the study has some challenges in making this data novel enough for publication and more detailed analysis is required to increase its novelty.

  1. From my point of view the main message of the manuscript was confusing - Is the focus here is to identify the Genexus pipeline as a candidate for a diagnostics pipeline in clinical laboratories or is it focusing on the 9% of samples that failed (and why they failed)?  If it is the former, then this should be clarified/confirmed in the text and data should be included on limit of sensitivity, inter/intra run variation and implications for difficult samples?  If it is the latter, more information regarding these samples (provenance/QC) should be included.

Authors’ response: We totally agree with the reviewer and we now clarified the aim of our study that was to report on the feasibility of using an NGS platform able to get fast results to identify Delta and Omicron SARS-CoV-2 variants, despite very low viral loads detected in different bioresources. We previously published on the diagnostic performance of the Genexus pipeline on specimens with sufficient quantity of viral RNA (Ref. 35: Hofman P, Bordone O, Chamorey E, et al. Setting-Up a Rapid SARS-CoV-2 Genome Assessment by Next-Generation Sequencing in an Academic Hospital Center (LPCE, Louis Pasteur Hospital, Nice, France). Front Med (Lausanne) 2021; 8: 730577).

Moreover, the quality check of the sequencing was performed with the “Ion AmpliSeq SARS-CoV-2-LowTiter Research Assay (Thermo Fisher Scientific), a new Assay specially designed for ≤ 200 copies. This was now added in the Manuscript.

  1. Was additional phylogenetic and variant analysis performed upon all samples other than those included in the pipeline?  

Authors’ response: We thank the reviewer for these comments. As described in the Materials and Methods section, the fastq files were quality filtered and reads mapped with the SARS-CoV-2-Pangolin plugin (https://cov-lineages.org/resources/pangolin.html) and “COVID19AnnotateSnpEff” automatic plugin (Thermo Fisher Scientific). In addition, we also used the Nextclade tool (https://clades.nextstrain.org).

We modified accordingly the section.

Were omicron minority variants picked up earlier than the peak - if so, at what time and could these have been used to predict a new wave as argued in both the introduction and discussion?

Authors’ response: We thank the reviewer for these important comments. As showed in Figure 1, omicron minority variants were picked up earlier than the peak, 15 days earlier. Thus, this could have been used to predict a new wave, as discussed in the Discussion section.  

  1. How were these results validated – whilst I appreciate that the authors have stated that no orthogonal analysis was performed (L190) this significantly narrows the number of conclusions that can be made on this data?  What are the implications for contamination using this pipeline, for instance?

Authors’ response: We totally agree with the reviewer that not using orthogonal methods narrows the number of conclusions. However, because of limited budget, we didn’t have the possibility of performing these methods unfortunately. Moreover, we would like to emphasize that the Genexus pipeline is already in routine use in our laboratory, and that risk management is carried out as part of our quality approach according to the ISO 15189 standard. Additionally, we are performing intra- and inter-laboratory controls regularly.

  1. Other than simply Delta and Omicron outputs from the assay, were any additional data available to identify other novel variants.  As stated on L230 this pipeline is amplicon based and therefore is sensitive to novel variants and amplicon drop-outs/revisions.  How did that translate across all samples. What % of genomes were identified in samples using this method?

Authors’ response: The 30 kb viral genome is 100% covered by the amplicons of the panel and the assay used. While these are able to identify novel variants, however, in our study, we did not identify novel variants.

  1. How do these results relate to the clinical outcome of the cases?  Whilst the paper acknowledges its small sample set, how relevant are these results to the larger picture in France (rather than just the South eastern region)?

Authors’ response: We thank the reviewer for this  important question, that we addressed now in the revised Manuscript. The detection of SARS-CoV-2 using RT-PCR can demonstrate high variability of sensitivity in various diagnostic specimens (bronchoalveolar lavage, double naso/oropharyngeal swabs, nasopharyngeal swabs, saliva, and oropharyngeal swabs) as reported elsewhere (Khiabani et al. Am J Infect Control 2021;49(9):1165-1176). A high Ct PCR, i.e. a very low viral load on a specific sample does not systematically predict the absence of an active viral infection. Thus, being able to identify a viral strain on a not very sensitive specimen can be of interest for some patients, at risk of severe SARS-CoV-2 infection, and even more if new variants are emerging. In the era of active treatments like antivirals and monoclonal antibodies that have a variable efficiency according to the viral strains, being able to identify any variant could have clinical consequences in terms of treatment strategies. Additionally, our findings extend far beyond local problems and are significant for epidemiological purposes.

  1. Relating to point 1, how did the authors identify samples as ‘certainly due to degraded and low viral load’ (L95)?  What were these observations based on?  This should be a minimal requirement for the manuscript as its main focus is on those samples that are low level/undetectable by other methods.  

Authors’ response: We thank the reviewer for these comments. Quantitative RNA data can be detected by RT-qPCR (copy number) and Idylla (Ct viral load). The quality of the amplified RNAs can be checked during sequencing by analyzing the % N (undefined nucleotide). We now modified accordingly the manuscript.

Could these be used to predict problem samples before sequencing? 

Authors’ response: We could analyze the % N only after sequencing as stated in the Materials and methods section.

Reviewer 3 Report

This is a potentially interesting article demonstrating the feasibility of sequencing samples with low viral load, and the possible difference (or lack of it) in patients infected with different strains. However, many questions arise.

1. Authors say that “screening by full genome sequencing of SARS-CoV-2 is performed in only 12 to 40% of positive cases, depending on the national recommendations,”

It will be very interesting to hear in which countries 12-40% of all samples are sequenced. For France, for example, this would mean thousands of samples daily that would need to be sequenced genome wide every day.

2. “the RT-PCRs for screening pose a problem of specificity by amplifying three targeted genes, which does not allow for the rapid identification of new variants”

In general, PCR is not meant to identify new variants, this is what sequencing is for, and therefore the comparison is incorrect. PCR is suitable for the sensitive detection of viral RNA. Sequencing, by definition, serves to read the genome or part of it, look for mutations, and, accordingly, determine the variant. Also, there is not much difference whether one or three fragments of the genome are determined in PCR.

3. “This approach could allow a more discriminant screening strategy for genomic monitoring than conventional RT-PCR assays”

It is not entirely clear what the authors mean by this. That NGS sequencing is more sensitive than RT-PCR? For sample preparation, PCR is used to amplify genome fragments, and so PCR is by definition more sensitive, otherwise NGS will yield no reads.

4. “we analyzed by NGS 23 specimens identified as negative with Idylla RT-PCR SARS-CoV-2. Among these, 39% (9/23) were negative with NGS; however, 61% (14/23) were positive for SARS-CoV-2 giving a high rate of false-negative results. This preliminary result must not preclude the use of RT-PCR assays for SARS-CoV-2 screening”

The question is, how can NGS sequencing, for which PCR is used for sample preparation, be more sensitive than high-quality PCR test systems?

 In addition, very few samples were used for this type of research, and there is no information about comparing the cost of similar studies (PCR and NGS), and in the end, the purpose of the article should be clearly stated.

In addition, it could also be worth mentioning the comparison of solutions from TFS and other manufacturers - comparison of price, data quality, speed, etc.

Author Response

Reviewer #3:

This is a potentially interesting article demonstrating the feasibility of sequencing samples with low viral load, and the possible difference (or lack of it) in patients infected with different strains. However, many questions arise.

  1. Authors say that “screening by full genome sequencing of SARS-CoV-2 is performed in only 12 to 40% of positive cases, depending on the national recommendations,”

It will be very interesting to hear in which countries 12-40% of all samples are sequenced. For France, for example, this would mean thousands of samples daily that would need to be sequenced genome wide every day.

Authors’ response: We totally agree with the reviewer that these % are overestimated. According to the genomic sequences shared via GISAID, the global data science initiative, (https://gisaid.org/submission-tracker-global/), the EMERGEN consortium (Consortium for Surveillance and Research on EMERgent Pathogen Infections via Microbial GENomics), coordinated by Santé publique France and ANRS | Emerging Infectious Diseases, full genome sequencing in France is performed in approximately 1.5% of positive cases. We now modified accordingly the Manuscript (page 2, line 54).

  1. “the RT-PCRs for screening pose a problem of specificity by amplifying three targeted genes, which does not allow for the rapid identification of new variants”

In general, PCR is not meant to identify new variants, this is what sequencing is for, and therefore the comparison is incorrect. PCR is suitable for the sensitive detection of viral RNA. Sequencing, by definition, serves to read the genome or part of it, look for mutations, and, accordingly, determine the variant. Also, there is not much difference whether one or three fragments of the genome are determined in PCR.

Authors’ response: We totally agree with the reviewer and we now modified accordingly the Manuscript (page 2, lines 60-66).

  1. “This approach could allow a more discriminant screening strategy for genomic monitoring than conventional RT-PCR assays”

It is not entirely clear what the authors mean by this. That NGS sequencing is more sensitive than RT-PCR? For sample preparation, PCR is used to amplify genome fragments, and so PCR is by definition more sensitive, otherwise NGS will yield no reads.

Authors’ response: We totally agree with the reviewer that this was very confusing and we now modified accordingly the Manuscript (page 4, lines 112-116).

  1. “we analyzed by NGS 23 specimens identified as negative with Idylla RT-PCR SARS-CoV-2. Among these, 39% (9/23) were negative with NGS; however, 61% (14/23) were positive for SARS-CoV-2 giving a high rate of false-negative results. This preliminary result must not preclude the use of RT-PCR assays for SARS-CoV-2 screening”

The question is, how can NGS sequencing, for which PCR is used for sample preparation, be more sensitive than high-quality PCR test systems?

Authors’ response: Other research studies showed similar results. Khan and Cheung [Khan K. A., Cheung P. Presence of mismatches between diagnostic PCR assays and coronavirus SARS-CoV-2 genome. Royal Society Open Science . 2020;7(6)] noted the presence of mismatches when comparing SARS-CoV-2 between RT-qPCR and sequencing data. Elaswad and Fawzy [Elaswad A., Fawzy M. Mutations in animal SARS-CoV-2 induce mismatches with the diagnostic PCR assays. Pathogens . 2021;10(3):p. 371] also found this to be the case when comparing RT-qPCR assays with available SARS-CoV-2 genomes isolated from animals. Similarly, Hoang et al. [Hoang P., Nguyen H., Tran H., et al. Missed detections of influenza A (H1)pdm09 by real-time RT-PCR assay due to haemagglutinin sequence mutation, december 2017 to march 2018, northern Viet Nam. Western Pacific Surveillance and Response Journal . 2019;10(1):32–38] noted missed detection with RT-qPCR assays for influenza A (H1) when compared with sequencing. Although these studies add some concern, it does appear strategic deployment of both NGS and RT-qPCR technologies for the discovery and monitoring of emerging SARS-CoV-2 mutations is likely to advance better strategies for epidemiological characteristics.

In addition, very few samples were used for this type of research, and there is no information about comparing the cost of similar studies (PCR and NGS), and in the end, the purpose of the article should be clearly stated. In addition, it could also be worth mentioning the comparison of solutions from TFS and other manufacturers - comparison of price, data quality, speed, etc.

Authors’ response: We agree with the reviewer and we now clarified the goal of our study that was to report on the feasibility of using an NGS platform to identify Delta and Omicron SARS-CoV-2 variants, despite very low viral loads. In addition, we now add the comparison of different sequencing solutions from TFS and other manufacturers

The Genexus sequencing has fewer gaps (<250 mean N per genome) than the Illumina and Oxford Nanopore Technology (ONT) platforms (>1250 mean N per genome) in a survey of 100,000 SARS-CoV-2 genome sequences [1]. Several research papers report high errors rates in ONT sequencing (5-8%) [2-4]. The Genexus system offers complete integration from library preparation to analysis in a single workflow, whereas ONT and Illumina have multiple instruments and 3rd party requirements. The Illumina approach needs separate instruments for automated liquid handling, thermocycler, plate centrifuge, qPCR system, and sequencer [5], whereas the ONT approach has 3rd party requirement for extraction, cDNA synthesis and bioinformatics analysis and is  a highly manual process [2]. In addition, the ONT system has the fastest turnaround time (~11 hours), the Illumina platform has the longest turnaround time (3-4 days), while the Genexus system lasts less than 24 hours. The Genexus platform has one touchpoint from library prep to analysis, the Illumina system requires large amount of hands-on time, and ONT has many steps (Supplementary Table 1) [6].

Supplementary Table 1. Comparison of different SARS-CoV-2 sequencing approaches.

Genexus Integrated Sequencer

Illumina NextSeq RUO COVID-Seq

Midnight ONT

Workflow TAT

1 working day

3 working days

1 working day

Total run time

24 hours

40 hours

11 hours

Hands-on time

5 minutes

3 hours

1 hour 10 minutes

Touchpoints

1

>80

>55

Finally, the Genexus system is an easy-to-use “plug and play” with no requirement for prior bioinformatics experience, and has an integrated software for sequencing analysis and reporting. The Illumina platform requires effort to automate, the need to develop and verify scripts for automation for different liquid handlers. For instance, the DRAGEN COVID lineage analysis workflow is complex with known limitations [7].

References

  1. Tegally H, San JE, Cotten M, et al. The evolving SARS-CoV-2 epidemic in Africa: Insights from rapidly expanding genomic surveillance. medRxiv 2022: 2022.04.17.22273906.
  2. Hourdel V, Kwasiborski A, Baliere C, et al. Rapid Genomic Characterization of SARS-CoV-2 by Direct Amplicon-Based Sequencing Through Comparison of MinION and Illumina iSeq100(TM) System. Front Microbiol 2020; 11: 571328.
  3. Bull RA, Adikari TN, Ferguson JM, et al. Analytical validity of nanopore sequencing for rapid SARS-CoV-2 genome analysis. Nat Commun 2020; 11(1): 6272.
  4. Genomic sequencing of SARS-CoV-2: a guide to implementation for maximum impact on public health. World Health Organization. Geneva, 2021.
  5. Bhoyar RC, Senthivel V, Jolly B, et al. An optimized, amplicon-based approach for sequencing of SARS-CoV-2 from patient samples using COVIDSeq assay on Illumina MiSeq sequencing platforms. STAR Protoc 2021; 2(3): 100755.
  6. Pembaur A, Sallard E, Weil PP, Ortelt J, Ahmad-Nejad P, Postberg J. Simplified point-of-care full SARS-CoV-2 genome sequencing using nanopore technology. medRxiv 2021: 2021.07.08.21260171.
  7. Lambisia AW, Mohammed KS, Makori TO, et al. Optimization of the SARS-CoV-2 ARTIC Network V4 Primers and Whole Genome Sequencing Protocol. Front Med (Lausanne) 2022; 9: 836728.

Round 2

Reviewer 1 Report

The revised version is now fine.  The authors have  made the necessary improvements.

Reviewer 2 Report

I believe that the corrections are sufficient to warrant the modified manuscript can be published.

Reviewer 3 Report

The article has no significant scientific significance but can be accepted in present form